# Enzyme Modifications of Red Deer Fat to Adjust Physicochemical Properties for Advanced Applications

**DOI:** 10.3390/molecules30153293

**Published:** 2025-08-06

**Authors:** Tereza Novotná, Jana Pavlačková, Robert Gál, Ladislav Šiška, Miroslav Fišera, Pavel Mokrejš

**Affiliations:** 1Department of Food Technology, Faculty of Technology, Tomas Bata University in Zlín, Vavrečkova 5669, 760 01 Zlín, Czech Republic; t2_novotna@utb.cz (T.N.); gal@utb.cz (R.G.); siska@utb.cz (L.Š.); 2Department of Fat, Surfactant and Cosmetics Technology, Faculty of Technology, Tomas Bata University in Zlín, Vavrečkova 5669, 760 01 Zlín, Czech Republic; pavlackova@utb.cz; 3Department of Food Analysis and Chemistry, Faculty of Technology, Tomas Bata University in Zlín, Vavrečkova 5669, 760 01 Zlín, Czech Republic; fisera@utb.cz; 4Department of Polymer Engineering, Faculty of Technology, Tomas Bata University in Zlín, Vavrečkova 5669, 760 01 Zlín, Czech Republic

**Keywords:** red deer fat, enzymatic lipid modification, degree of hydrolysis, melting temperature, crystallization temperature, functional groups, texture, color parameters, sustainability

## Abstract

Red deer fat makes up approximately 7–10% of the animal’s weight and is not currently used. Regarding sustainability in the food industry, it is desirable to look for opportunities for its processing and use, not only in the food industry. The aim of this study is the enzymatic modification of red deer fat, leading to modification of its physicochemical properties, and the study of changes in phase transitions of modified fat, its structure, color, and texture. Hydrolysis was performed using sn-1,3-specific lipase at different water concentrations (10–30%) and reaction times (2–6 h). The results showed that there was a significant decrease in melting and crystallization temperatures with an increasing degree of hydrolysis, which was confirmed by differential scanning calorimetry. FTIR spectra revealed a decrease in the intensity of the ester bonds, indicating cleavage of triacylglycerols. Texture analysis of the modified fats confirmed a decrease in hardness of up to 50% and an increase in spreadability. The color parameter values remained within an acceptable range. The results show that enzymatic modification is an effective tool for targeted modification of red deer fat properties, and this expands the possibilities of its application in cosmetic matrices and food applications as functional lipids.

## 1. Introduction

The red deer (*Cervus elaphus*) is one of the most widespread cervid species, and its habitat is not limited to Europe, but also extends into Asia and North America [1]. In European countries, the meat of this animal has long been consumed and, thanks to its favorable nutritional properties, deer farming is becoming more frequent. This type of farming is most common in New Zealand, with smaller farms also existing in Europe, North America, Australia, and tropical areas such as Mauritius [2,3].

Texture and taste are key factors that consumers often consider most important when evaluating the food quality of meat. Different consumer groups have different preferences for these characteristics, which have implications for the market for both domestic meat and venison. In Europe, venison meat is associated with the autumn hunting season and slaughter activities and is valued for its characteristic “wild” taste. By contrast, in other countries, such as the USA, the “wild” taste of venison is not favored and is considered undesirable [4]. From a nutritional point of view, it is a quality meat as it contains a high proportion of polyunsaturated fatty acids (PUFAs) and has a satisfactory ratio of omega-6 to omega-3 fatty acids (n6/n3) [1,5]. In ruminants, especially venison, the influence of the fatty acid composition of the meat is diverse, as fatty acids are hydrogenated by microorganisms in the rumen [4,6]. Disadvantages include its higher cholesterol content compared to beef [5].

The protein content of deer meat varies between 19.3% and 23.6%, while the fat content ranges from 1.7% to 4.6% by weight [7]. This type of meat is becoming increasingly appealing to consumers, particularly due to its low fat content and high polar lipid content, making it one of the most nutritious meats [8].

Polar lipids, particularly phospholipids, are biologically active components that contribute not only to the nutritional value of meat but also to human health, due to their anti-inflammatory and cardioprotective properties [6]. They also enhance key sensory attributes such as juiciness and flavor intensity—factors that are strongly associated with consumer preference. The presence of these lipids improves palatability, which can positively influence the perception of lean game meats like venison, often seen as a natural and sustainable food source [4,9].

Venison contains, on average, 50–80% less fat than meat from livestock, and the specific composition of deer meat distinguishes it from conventional beef or pork [10]. However, some consumers may be concerned about the safety and quality of this meat, particularly with regard to microbiological contamination and heavy metal content [2]. Research shows that the content of heavy metals in meat depends mainly on the pollution of the environment in which the animals live, with deer meat containing the least heavy metals compared to other game species [11]. Microbiological contamination can be reduced by proper hunting techniques, observing all hygiene standards, and proper heat treatment of meat [12].

Triacylglycerols are the main lipid component of animal fats, accounting for more than 95% of their total content. The fat in deer meat is characterized by its low unsaturated and higher saturated fatty acid content, making it a very hard fat [4]. The venison tallow is mostly unused, even though it shows high stability. However, this type of fat shows considerable potential for use in various industries, particularly in the food, cosmetics, and pharmaceutical industries [13]. Despite this potential, deer fat is often treated as a by-product and is discarded or underutilized, even though it contains essential fatty acids and phospholipids with possible applications in food, cosmetics, and pharmaceuticals (e.g., as emulsifiers or bases for skin and body cosmeceuticals) [9]. Enzymatic hydrolysis of animal fats is considered a mild and environmentally friendly “green technology” that enables their transformation into functional, value-added ingredients. This approach supports circular economy principles, reduces waste, and may also allow for the recovery of other valuable biomolecules such as fatty acids, proteins, or collagen from meat by-products [14]. Therefore, the valorization of deer fat through enzymatic processing is not only a way to explore its functional potential but is also a contribution to Sustainable Development Goal 12 (Responsible Consumption and Production) by promoting efficient by-product utilization and a reduction in the environmental impact of the food industry [15].

The fat is mainly used to influence the organoleptic properties of products. It can act as a texturizer, lubricant, carrier of flavor, or other functional or active fat-soluble substances [16,17]. The textural and rheological properties of fat are closely related to the structure of its crystal networks [18]. Through appropriate fat modification, it is possible to obtain matrices with a wide range of applications. The most commonly used method of fat modification is chemical modification, but biological methods using enzymes are gaining popularity [19].

Lipases (EC 3.1.1.3) are key hydrolytic enzymes that specialize in the cleavage of ester bonds in triacylglycerols. They are characterized by high chemo-, regio-, and often stereoselectivity, making them a versatile tool for lipid modification in the food, pharmaceutical, and chemical industries [20,21]. They occur naturally in microorganisms, plants, and animals, with the most commonly used industrial variants coming from fungi (e.g., *Rhizomucor miehei*, *Candida rugosa*, and *Thermomyces lanuginosus*) [22]. Modern production involves the recombinant production of lipases in microbial hosts, thus ensuring higher yields and stability. Thanks to advanced protein engineering methods such as directed evolution or rational design, it is now possible to target changes in substrate specificity and increase enzyme stability to temperature, pH, and organic solvents [20,23].

Enzymatic modification of fats is a gentle and highly selective method for modifying their physicochemical and nutritional properties [24]. One of the key techniques is partial enzymatic hydrolysis using specific lipases, in particular sn-1,3-specific lipases, which selectively cleave ester bonds at the extreme positions of the glycerol skeleton of triacylglycerols [25]. This type of enzyme allows for controlled modification of fat structures without disrupting the central sn-2 position, which often carries bioactive fatty acids [25,26].

The use of sn-1,3-lipases is particularly effective in the modification of animal fats, where a significant change in texture, softening of the product, and improvement of its processing properties occur [24,26,27]. In addition, partial hydrolysis lowers the melting point and allows the production of functional lipids with better spreadability and more easily absorbable formations. Enzymatic modification thus offers an environmentally friendly alternative to conventional chemical methods, without the formation of trans fatty acids [28].

The transition of fat, such as tallow, from a liquid to a solid form is called crystallization. This process depends on the fat’s temperature, cooling rate, and fatty acid composition. Upon cooling, the fatty acid molecules arrange themselves into crystalline structures, which can take various forms [29]. The main forms of tallow crystals include the beta (β) form, which is stable and solid, and the beta-prime (β′) form, which forms on faster cooling and has a finer structure. The alpha (α) form is the least stable and forms on rapid cooling, with the crystals being brittle and gradually changing to more stable forms [30,31]. Additionally, a combination of the two forms can occur during crystallization, which affects the properties of the tallow, such as texture, stability, and melting temperature [32]. To control the crystallization and melting of samples, thermal analyses are used, including differential scanning calorimetry (DSC), which allows the study of samples with varying heating and cooling rates over a wide temperature range (typically from −175 °C to 700 °C) [33,34]. The melting temperature of tallow ranges from 43 to 47 °C [35].

Fourier transform infrared (FTIR) spectroscopy is widely used to identify and monitor functional groups in the structure of fats and oils. This method is a rapid and non-invasive analytical technique for the detection of characteristic bond vibrations, such as -C=O (carbonyl group), -CH_2_, -CH_3_, and -OH, which are important for the evaluation of the chemical composition, oxidation, and possible degradation of fat components [36]. FTIR can also be used to monitor changes due to technological treatments, for example, during interesterification or lipid oxidation [37]. The spectral region around 1740 cm^−1^ is typical for fatty acid esters, while the bands in the region 3000–2800 cm^−1^ correspond to the valence vibrations of C-H bonds in alkyls [38,39]. The advantage of FTIR is not only its speed and low demands on sample preparation, but also the possibility of using it with chemometric methods for quantitative evaluation of lipid components [40].

Other properties monitored for modified fats are color parameters and texture properties. Color plays a key role, for example, in cosmetic products and the customer’s first impression, and often influences the perception of the quality, efficacy, and freshness of the product. Color changes can indicate chemical degradation, oxidation of active ingredients, or interactions between components; therefore, color monitoring is important for quality control and product stability [41]. For the unbiased assessment of color, spectrophotometers and measurements in the CIE L*a*b* system are commonly used to provide a quantitative description of color shades [42]. Color is particularly important in natural products and emulsions where undesirable changes may occur during storage [43,44]. Texture properties can be significantly affected by enzymatic modification. This process alters the composition and arrangement of triacylglycerols, leading to changes in the crystalline structure of the fat and thus modifying its consistency, plasticity, and spreadability [45,46].

The main objectives of the work were (1) to evaluate the influence of process factors on the partial enzymatic hydrolysis of deer tallow, (2) to analyze the structural changes in the lipid profile of the modified fat, (3) to assess the influence of partial enzymatic hydrolysis on the phase transitions of the modified fat, and (4) to propose suitable conditions of enzymatic modification of deer tallow for applications in cosmetic and food matrices.

## 2. Results and Discussion

### 2.1. Composition of Fatty Acids in Deer Tallow

The composition of fatty acids in the red deer (*Cervus elaphus*) fat is given in Table 1.

The results of the analysis showed that deer tallow contains a high proportion of saturated fatty acids (SFAs; 66.64%), with palmitic acid (35.55%) and stearic acid (20.90%) dominating. The content of monounsaturated fatty acids (MUFAs) is relatively low (11.05%), while polyunsaturated fatty acids (PUFAs) account for 21.55%, with linoleic acid being prominent (21.13%). This profile is typical of ruminant fats and contributes to the firm texture and higher melting point. The measured values correspond to those available in the literature for red deer fat [1,4,5,47,48], confirming the stability of tallow composition across geographic areas and range conditions. Slight differences may be due to factors such as age, the forage base, and seasonal variation. An additional unidentified peak (“Unknown”) was detected between C14:0 and C14:1 in Table 1. This signal most likely represents a less common fatty acid with a similar carbon chain length—such as an isomer, a trans-form, or a methylated derivative—not covered by the FAME standard used. Such unidentified minor peaks are common in the GC-FID analysis of natural fats, especially when the sample contains trace levels of rare or structurally similar fatty acids [49,50].

### 2.2. Degree of Hydrolysis and Acid Value

The process conditions for the enzyme modification of deer tallow and the results of the acid value and degree of hydrolysis are recorded in Table 2.

The AV expresses the amount of free fatty acids released during enzymatic hydrolysis and serves as a rapid indicator of its progress. The results clearly show that the AV increases with the increasing water content and prolonged reaction time, confirming the efficient cleavage of triacylglycerols. The highest AV (115.54 mg KOH/g) was observed in the sample with the maximum DH (49.07%). On the contrary, the AV remained very low in the blind experiment, confirming that no significant hydrolysis occurs in the absence of the enzyme. Thus, an increased AV is an indicator of successful tallow modification.

The DH was subject to analysis of variance. The results of this analysis are presented in Table 3. Figure 1 shows the relationship between the dependent and independent variables.

The regression analysis results of the hydrolysis degree did not show statistical significance for any of the observed factors (*p* ≤ 0.05). Figure 1 shows that the lowest DH (<32%) was observed at a low water content (up to 10%) and a short reaction time (2 h); the highest DH (>48%) was obtained at a high water content (30%) and a long reaction time (6 h). The DH increased with increasing time and water content, confirming the positive effect of these factors on the efficiency of enzymatic hydrolysis. The results were compared with the study of Teng et al. (2010), which deals with the enzymatic hydrolysis of chicken fat. In both cases, a positive effect of hydrolysis duration on increasing the degree of hydrolysis was confirmed. The highest DH in our experiment (49.07%) corresponds to the values obtained in this study. The differences in the reaction conditions are mainly due to the different types of fat and the system used, but the results confirm a similar trend in the influence of technological factors on the hydrolysis process [51]. In the study by Alahmad et al. (2023), which investigated the effect of various factors on the DH, similar results to this work were found when evaluating the effect of reaction time. There was a gradual increase in the degree of hydrolysis with an increasing reaction time [52].

The microbial sn-1,3-specific lipase used in this study exhibits high substrate and position specificity and catalyzes the hydrolysis of ester bonds at the sn-1 and sn-3 positions. Due to this selectivity, it is highly unlikely that significant hydrolysis of other lipid classes, such as phospholipids, which typically require different types of enzymes, would occur [14,27].

### 2.3. Vibrational Characterization of Functional Groups

Five characteristic peaks were selected from the obtained FTIR records. The molecular action for each peak is recorded in Figure 2. Wavenumbers of these peaks with reference values [53,54,55] and their absorbances are recorded in Table 4.

According to Figure 3, there are clear differences in the intensity of individual peaks, especially in the region around 720, 1170, 1740, 2850, and 2920 cm^−1^. Table 4 shows the absorbance values of selected characteristic bands of FTIR spectra for pure red deer fat, the control sample (experiment 10), and samples after partial enzymatic hydrolysis (experiments 1–9). The main wavenumbers monitored were approximately 720 cm^−1^ (=C-H (cis)), 1170 cm^−1^ (C-O stretch), 1740 cm^−1^ (C=O stretch), 2850 cm^−1^ (C-H methylene stretch symmetric), and 2920 cm^−1^ (C-H methylene stretch asymmetric).

The analysis mainly focused on the intensity of the peak at 1739 cm^−1^, which corresponded to the C=O stretch ester bond. This peak showed the most significant change in intensity between samples. The highest intensity was found in the purified fat and in sample 10 *, where no enzyme was used in the reaction. Conversely, the lowest intensity was observed for sample 9, which also showed the highest DH.

Comparing the intensity of this peak with different degrees of hydrolysis of the samples, it was found that the intensity of the band at 1739 cm^−1^ decreased with the increasing DH. This trend is indicative of ester bond cleavage in triacylglycerols and the formation of free fatty acids and mono- or diacylglycerols. Similar results were also observed in our earlier study aimed at monitoring the intensity of individual peaks in hydrolyzed deer tallow and their comparison with the DH [56]. A further decrease in intensity was observed at 1173 cm^−1^, corresponding to C-O stretch vibrations, which also confirms ester bond cleavage. In contrast, the band at 721 cm^−1^, corresponding to the =C-H cis bond, remained relatively stable, indicating that the geometry of the unsaturated fatty acids was not significantly affected by hydrolysis.

In summary, the results of FTIR analysis confirm the structural changes of lipids due to enzymatic hydrolysis, with the most significant differences observed in the region of ester bonds. These results are also in agreement with previous studies [51,57].

### 2.4. Thermal Properties

The thermal properties of the analyzed samples were evaluated by DSC, which allows detailed observation of the changes during phase transitions, especially in the melting region, the crystalline structure of the fats, and the changes that occur due to enzymatic modification. The course of the DSC analysis for sample 9 and the purified fat is shown in Figure 3, where the differences in the thermal profiles of the individual samples are evident. A summary of the measured melting and crystallization temperatures is given in Table 5. Both DSC and open capillary melting point determination evaluated the thermal properties of the samples. Both methods made it possible to observe changes in the phase transitions of the fat samples after enzymatic treatment.

From the results of DSC analyses, it is clear that the samples subjected to enzymatic hydrolysis showed a decrease in melting temperature compared to purified tallow. This decrease in the melting point is due to the change in the molecular structure of the lipids, in particular the increased proportion of free fatty acids, which disrupts the regular crystalline structure. The most significant decrease in melting point was observed for sample 9, which also had the highest degree of hydrolysis. The melting point values measured by the open capillary method were generally slightly higher than those obtained by the DSC method, which is a common phenomenon, since the capillary method detects the temperature of the complete dissolution of the sample, while DSC registers the start of melting (the endothermic effect). This result is consistent with the observations of other studies, which also report that enzymatic hydrolysis causes disruption of triacylglycerol molecules, thereby lowering the melting point [31,58,59]. The melting temperature results confirm that enzymatic hydrolysis has a significant effect on the physicochemical properties of fats and can be used for targeted modification of their functions.

Important changes were also observed in the crystallization temperatures. The crystallization temperature was reduced in the hydrolyzed samples compared to the original purified fat. This phenomenon is due to the altered size and polarity of the molecules after hydrolysis, which reduces the ability of the fats to form a stable crystal lattice at higher temperatures. The decrease in crystallization temperature suggests that fats tend to remain longer in the liquid state after hydrolysis and only crystallize at lower temperatures. This effect may be technologically advantageous, for example, in the manufacturing of products requiring a softer structure during storage.

Variance analysis was performed to evaluate the results achieved. A summary of the results is presented in Table 6. Figure 4 shows the relationship between the dependent and independent variables using a contour graph.

The results of the regression analysis for melting and crystallization temperature showed no statistical significance for any of the observed factors (*p* < 0.05). The contour graph in Figure 4a shows the effect of the water amount and the reaction time on the melting temperature of the modified tallow. With a higher water content and a prolonged hydrolysis time, a trend of a decreasing melting point is observed, confirming the efficiency of the enzymatic cleavage of triacylglycerols. The lowest melting temperatures (<35 °C) were achieved under conditions of maximum hydrolysis, which improves the plasticity of the fat for industrial use. The graph in Figure 4b shows the effect of process factors on the crystallization temperature of tallow. Similar to the melting point, a higher water content and a longer hydrolysis time decrease the crystallization temperature, making the fat behave more like a softer lipid matrix. The lowest crystallization temperatures (<31 °C) promote longer retention of the fat in the liquid state, which is advantageous in applications requiring better spreadability.

### 2.5. Texture Properties

Partial enzymatic hydrolysis led to changes in the texture properties of the evaluated samples. The textural parameters were monitored using the spreadability test, which allows the assessment of the mechanical properties of the fat matrix, in particular, hardness, spreadability, stickiness, and adhesiveness. The measured values of each parameter are presented in Table 7.

Enzymatic hydrolysis of deer tallow led to significant changes in textural parameters, see contour graphs in Figure 5. The most significant changes were observed in the hardness parameter, where a significant decrease was observed in samples with a higher degree of hydrolysis. Sample 9, with the highest DH value (49.07%), showed the lowest hardness (25.8 N), while sample 4, with a lower DH, had a value of over 52 N. This trend confirms that the cleavage of triacylglycerols disrupts the crystalline structure of the fat and contributes to its softening, which has been previously described in other studies dealing with the modification of animal fats, e.g., in the study by Zou et al. (2023) for beef tallow [45].

The spreadability parameter showed a similar trend. Samples with a higher degree of hydrolysis were characterized by lower spreading resistance values, which may be advantageous, for example, in the formulation of cosmetic products where easy application is a key factor [46]. For the most hydrolyzed sample (Ex. No. 9), the spreadability value was only 18.1 N·m, which is significantly lower than that of sample No. 4 (40.9 N·m), confirming the effect of enzymatic treatment on fat processability.

In the case of stickiness and adhesiveness, the differences were even more pronounced. Sample 9 achieved extremely high values of stickiness (4.64 N), while purified fat without enzymatic treatment showed a value of approximately 1.60 N. These results suggest that hydrolysis not only disrupts the crystalline structures but also results in the formation of components (e.g., monoacylglycerols) that increase the stickiness and adhesion of the material. These changes may be desirable, for example, in the application of ointments or topical emulsions.

Statistical analysis presented in Table 8 showed that of all the textural parameters, only the dependence of stickiness on the amount of added water was statistically significant (*p* ≤ 0.05). The other parameters did not show statistically significant differences. The lower values of hardness with spreadability at a low water content and a short reaction time could be due to the cracking of the samples during the course of measurement.

The results of this study are in line with the findings of Nusantoro et al. (2016), who also observed a significant decrease in hardness and an increase in stickiness during enzymatic interesterification of lauric fat mixtures depending on the degree of restructuring of the triacylglycerol structure [59]. Similarly, Pruchnik et al. (2022) showed that lipid modification with sterols or enzymes affects not only the melting point but also the plasticity and mechanical properties of the resulting fat [58].

### 2.6. Color Parameters

Color is one of the essential parameters that consumers register, and although often underestimated, it plays a key role in the sensory perception of product quality. The color parameters of samples prepared after partial enzymatic hydrolysis were evaluated by colorimetric analysis. Based on the measured values, the overall color deviation (∆E*) was calculated to assess the color changes of the individual samples. Purified fat was used as a reference standard for color assessment. The measured results are presented in Table 9.

The results show that the highest overall color deviation (ΔE* = 7.61) was observed for sample 7, while other, less hydrolyzed samples had ΔE* lower than 3. In general, it is reported that at ΔE* higher than 3, a color change becomes visually apparent [60]. The most significant color changes were correlated with higher b* (yellow hue), which may be due to an increase in free fatty acid content or incipient oxidation.

Although samples 8 and 9 were processed at the same water content (30%) as sample 7, they showed lower color deviation (ΔE*), even though they had a higher degree of hydrolysis (41.5% and 49.1% compared to 33.9%). One possible explanation is that sample 7 underwent only partial hydrolysis of triacylglycerols, which could have led to a higher proportion of intermediates such as monoacylglycerols and free fatty acids. These substances may be more susceptible to oxidation or emulsification, which could lead to increased turbidity or optical instability. In addition, the short reaction time in sample 7 could have caused incomplete phase separation or higher water retention, which would further contribute to color instability. In contrast, samples 8 and 9 likely underwent more advanced enzymatic conversion, with better chemical stability and more efficient removal of residual water, thereby minimizing color changes. Other studies have described similar phenomena, where more pronounced color fluctuations occurred in partially hydrolyzed fat systems [43,46].

Conversely, the lowest ΔE* (<3) was observed for samples 1, 3, 5, and 9, which also had lower or intermediate values of hydrolysis degree. Thus, for these samples, it can be assumed that the color properties were preserved, which may be particularly advantageous in applications where the appearance of the fat plays a role, such as in cosmetic formulations or some food products.

The parameter L* (lightness) remained relatively high in all samples (mostly above 85), indicating that even the more intensive enzymatic reaction did not significantly change the optical lightness of the fat. Similar trends were also described in our study, where changes in ΔE* after enzymatic treatment of the deer tallow were directly related to the degree of hydrolysis, with the lightness and saturation of the yellow hue being influenced mainly by the composition of the products formed and the degree of oxidation [56].

### 2.7. The Relevance for Practice and the Limitations of the Study

#### 2.7.1. Optimal Conditions for Enzymatic Modification and Utilization of Modified Fats

Red deer fat modified with 2.0% enzyme with 30% water addition and a hydrolysis time of 6 h had the highest DH (49.07%). The texture properties of this sample are most suitable for cosmetic applications where low hardness and good spreadability of the product are expected. In case of further modifications, such as removal of free fatty acids, it would be possible to obtain a mixture of monoacylglycerols (MAGs) and diacylglycerols (DAGs), which could also find application in the food industry as a functional lipid component.

#### 2.7.2. Limitations of the Study

During enzymatic hydrolysis, almost 50% of all fatty acids were released, which represents a significant degree of transformation of the triacylglycerol structure. Further increases in the amount of water added would probably not lead to a significant increase in the DH because the sn-1,3-specific lipase used cleaves ester bonds only at the terminal positions of the glycerol. Moreover, a higher proportion of water would make the subsequent technological processing more difficult, especially during the separation of the water phase and the drying of the product, which could negatively affect the efficiency and stability of the resulting fat.

#### 2.7.3. Future Perspectives

The processing of fatty by-products from red deer (*Cervus elaphus*) represents an opportunity to reduce the costs associated with the disposal of animal by-products, which are often considered waste. By converting these raw materials into value-added products such as modified lipids suitable for use in the cosmetics, food, or pharmaceutical industries, the sustainability and economic efficiency of game meat processing can be significantly improved. The use of natural raw materials corresponds to current trends in sustainable technologies and the circular bioeconomy. The enzymatic modification used in this study represents an environmentally friendly alternative to traditional chemical methods. In particular, the use of microbial lipases allows reactions to be carried out under mild conditions without the need for aggressive solvents or catalysts. This opens up the possibility of developing so-called “green technologies” for the treatment of fats with low environmental impact.

In the future, further improvement of separation and production processes can be considered in order to produce mono- and diacylglycerols with higher added value and wider application potential. A similar approach could also be applied to other animal fats, such as beef, pork, or fats from other game species, which are commonly available but technologically less utilized. By targeted enzymatic treatment of these raw materials, their valorization and conversion into functional lipid components that are useful across the food, cosmetic, and pharmaceutical industries can be achieved.

## 3. Materials and Methods

For partial enzymatic hydrolysis, the microbial enzyme Lipex^®^ Evity 200 L (Novozymes, Copenhagen, Denmark), which belongs to the group of sn-1,3-specific lipases, was chosen. This enzyme selectively hydrolyzes ester bonds at the terminal positions of the glycerol skeleton of triacylglycerols, thus allowing targeted modification of the fat structure without interfering with the central sn-2 position.

The following equipment was utilized: a Braher P22/82 meat mincer (Braher, San Sebastian, Spain), a Memmert chamber dryer (Memmert GmbH + Co. KG, Büchenbach, Germany), Kern 440-47 electronic scales (KERN & SOHN GmbH, Balingen, Germany), a compact FT-IR spectrometer ALPHA II BRUKER (BRUKER, Billerica, Massachusetts, USA), a TA.XTplus Texture Analyser (Stable Micro Systems Ltd., Godalming, UK), an UltraScan VIS Spectrophotometer HunterLab (HunterLab, Reston, VA, USA), a heating plate (Schott Geräte, Mainz, Germany), Thermal Analysis DSC 1 (Mettler Toledo, Greifensee, Switzerland), the Agilent 7820A GC System (Agilent Technologies, Santa Clara, CA, USA), the high performance microwave digestion system ETHOS ONE (MILESTONE, Sorisole, Italy).

### 3.1. Raw Material

Fat tissue from red deer (*Cervus elaphus*) was supplied in cooperation with Venison CZ Ltd. (Míškovice, Czech Republic), originating from farmed animals. The fat tissue was analyzed using conventional food methods, including water [61], protein [62], fat [63], and inorganic matter [64] content. The individual analyses were performed four times, and then the mean and standard deviation were calculated. To determine the acid value (AV) [65], saponification value (SV) [66], peroxide value (PV) [67], and iodine value (IV) [68], the tissue was purified, and pure fat was used for analysis. The water content of the starting material was 19.00 ± 1.93%. In the dry matter, the content of the individual components was in the following ranges: protein 5.23 ± 1.49%, fat 95.46 ± 1.37%, and inorganic matter 0.28 ± 0.12%. The ranges for pure fat were as follows: AV 2.04 ± 0.09 mg/g, SV 235.49 ± 6.41 mg/g, PV 7.2 ± 1.5 µval/g fat, and IV 20.8 ± 0.3 g/100 g.

### 3.2. Experiment Design and Statistical Analysis

Factorial experiments are commonly used to determine optimal conditions in industrial processes. Two-level factors are most commonly applied, but mixed-factor experiments are also commonly used [69]. Design of Experiments (DOE) allows for the analysis of how process factors (independent variables) affect dependent variables. Using the DOE methodology, it is possible to identify statistically significant factors that influence a process and then to optimize these processes [70].

In the experimental design, a multifactorial experiment method was used to influence the process conditions. A Taguchi design with two independent variables was used, with each variable studied at three levels. The values of each factor were chosen according to the results of previous studies [56,71,72]. The independent variables chosen for this experiment had three levels of factors: factor A—the amount of water (10, 20, 30%), and factor B—the reaction time (2, 4, 6 h). The dependent variables studied included the DH, the AV, texture properties, the melting point, and the crystallization temperature.

Analyses of modified fats were performed five times. Using Microsoft Office Excel 2016 (Microsoft, Denver, CO, USA), mean values were calculated from the measured values. Regression analysis was performed using the Minitab^®^ 18.1 statistical software (Fujitsu Ltd., Tokyo, Japan). Analysis of variance (ANOVA) was determined for the DH. Statistical significance of the factors was evaluated at the 95% significance level (*p* < 0.05); factors with a value <0.05 affected the process variables.

Graphical analysis of the data was performed using identical software to create contour graphs showing the relationships between the dependent and independent variables by applying Akima’s polynomial method of interpolation.

### 3.3. Preparation and Enzymatic Modification of Red Deer Fat

Figure 6 shows the processing steps for the preparation of enzymatically modified red deer fat.

The adipose tissue from red deer was supplied in the form of slices, which were cleaned with a knife from larger pieces of meat or other tissue and then cut into smaller pieces. These portions were homogenized with a meat slicer through a cutting system of two perforated plates (kidney-shaped holes, plate with 10 mm holes) with a double-sided knife between them. The homogenized raw material was vacuum packed and then shock frozen at −80.0 ± 1.0 °C for 12 h. The raw material was then stored at −18.0 ± 1.0 °C and thawed at 5.0 ± 1.0 °C prior to melting. The homogenized raw material was melted at 70.0 ± 5.0 °C in a chamber dryer without air circulation for 2 h. To obtain pure fat, the mixture was filtered through 4 layers of polyester filter fabric with a pore size of 0.5 mm; the yield was 44.3 ± 2.4%.

The enzymatic modification of the fat was carried out in a 250 mL laboratory reactor. First, 2.0% of the enzyme (by weight of fat) was added to the amount of water according to factor A (10, 20, 30%). Subsequently, molten fat (100.0 ± 0.1 g) was added, and the mixture was heated in a water bath at 50.0 ± 1.0 °C and stirred at a constant temperature at 450 ± 100 rpm for a time according to factor B (2; 4; 6 h). After the reaction time, the temperature was increased at a rate of dt/dτ 6 °C/min to 85.0 ± 0.5 °C for 5 min. During heating, silica gel (3 ± 1 mm particle size in an amount of 20 ± 10 g) was added, and after heating, the mixture was filtered using a Buchner funnel through Filpap KA-2 filter paper (Filpap, Ixelles, Belgium). The laboratory glassware had to be heated to about 100 °C before use.

### 3.4. Analytical Part

Fatty acid profile determination was performed according to ISO standards [73,74,75,76] with minor modifications. Samples (0.1 g of fat) were saponified in a microwave mineralizer with 5 mL of 2.8% methanolic KOH at 90 °C for 10 min, followed by methylation with 15 mL of methanolic acetyl chloride at 120 °C for 6 min. After cooling, the fatty acids were extracted with 10 mL of heptane, washed with saturated NaCl solution, and dried over sodium sulfate. Methyl ester analysis was performed on an HP-5 column (30 m × 0.32 mm × 0.25 µm) by GC-FID. The injector and detector were set at 250 °C, the split ratio was 1:50, and the carrier gas was hydrogen (3 mL/min). The temperature program was as follows: 140 °C (3.75 min), followed by 4 °C/min to 210 °C (6 min) and 2 °C/min to 235 °C (2 min). The total analysis time was 42 min. Fatty acids were identified by comparing retention times with the standard of fatty acids methyl esters 37 Component FAME Mix, Linoleic acid methyl isomer mix (Supelco, Bellefonte, PA, USA).

The dry matter content was determined by the gravimetric method. After burning the sample, the inorganic content of the dry matter was determined. Nitrogen was determined by the standard Kjeldahl method. After purification, the AV, SV, PV, and IV were determined for the pure fat according to food testing methods for oils and fats [65,66,67,68].

The acid value indicates the amount of free fatty acids in the sample. It is determined by ethanol titration of dissolved fat with a KOH solution to the neutral point. The saponification value expresses the total amount of alkali required to saponify the free and bound fatty acids. It is determined by titration after hydrolysis of the sample with excess KOH. The peroxide value indicates the amount of peroxide compounds formed during the primary oxidation of the fat. It is determined by reaction with potassium iodide, followed by titration of the released iodine with thiosulfate. The iodine value indicates the degree of unsaturation of the fat expressed as the number of grams of iodine absorbed by 100 g of oil or fat.

The degree of hydrolysis (DH) was determined using the American Oil Chemists’ Society methods [77]. The DH was calculated from the measured AV of pure fat (P) and partially hydrolyzed fat (H). The results were obtained using the following Equation (1):(1)$$DH\left(\%\right)=\frac{{AV}_{H}}{{SV}_{O}-{AV}_{O}}\times 100$$

For vibrational characterization of functional groups of the tested samples, Fourier transform infrared spectroscopy was used according to the Official Methods of Analysis [77]. The ALPHA II compact FTIR spectrometer was used for the analysis. The spectra were corrected against air spectra. The measurements were carried out in the range of 400–4000 cm^−1^ with a resolution of 8 cm^−1^, and 32 scans were taken. A reference air measurement was taken between each sample. The recorded spectra were expressed as absorbance values, and a graphical representation of the dependence of absorbance (au—arbitrary units) on wavenumbers (cm^−1^) was also plotted.

Differential scanning calorimetry was used to determine the melting and crystallization temperatures of the samples. The analysis was performed using a modified method [78], which has been used in similar types of matrices. The measurements were carried out on the DSC equipment. For the analysis, 5.0 ± 0.5 mg of the sample was weighed into special aluminum trays, which were then hermetically sealed. The samples were placed in the apparatus, heated to 70 °C at 5 °C/min, and held for 5 min. The samples were then cooled to 5 °C at 5 °C/min and held at this temperature for 5 min. The entire heating and cooling cycle was performed three times.

The melting point of the samples was determined in an open capillary according to the method of determination for animal and vegetable fats and oils [79]. The melting point in the open capillary corresponds to the temperature at which the fatty substance in the glass capillary starts to melt. The melting point corresponded to the moment when the sample had completely dissolved into a uniform, clear phase.

The texture properties were determined using a modified method applied to a similar type of matrix used in the study by Glibowski et al. (2008) [80]. This is a method designed for semi-rigid matrices using the TA.XT Plus Texture Analyzer. The measurements were carried out penetrometrically using an HDP/SR probe that was immersed in the sample to a depth of 2 mm at a speed of 1 mm/s. Prior to the actual measurement, the samples were melted 24 h in advance, poured into trays, and stored at 21.0 ± 2.0 °C for 2.0 ± 0.5 h. Subsequently, the trays with the samples were placed in a refrigerator (4.0 ± 2.0 °C) for 22 ± 1 h and cleaned before measurement. Each sample was measured four times. Based on the values obtained, the average values of texture parameters such as hardness, spreadability, stickiness, and adhesiveness were calculated.

The color measurements were carried out according to the methodology described in a previous study [81] using the UltraScan VIS spectrophotometer from HunterLab. Calibration of the device was performed on a black and white background. Subsequently, measurements of individual samples were carried out. Based on the measured values, the parameter ∆E* was calculated according to equation (2), where ∆L* represents the difference in the values of the L* component between the unhydrolyzed tallow and the sample, ∆a* is the difference in the values of the a* component between the unhydrolyzed tallow and the sample, and ∆b* is the difference in the values of the b* component between the unhydrolyzed tallow and the sample.(2)$$∆E^{*}=\sqrt{{∆L}^{*2}+{∆a}^{*2}+{∆b}^{*2}}$$

## 4. Conclusions

This study showed that partial enzymatic modification of deer fat by sn-1,3-specific lipase is an effective method for targeted changes in its physicochemical properties. There was a significant decrease in melting and crystallization temperatures with an increasing degree of hydrolysis, which DSC confirmed. FTIR analysis confirmed structural changes, mainly a decrease in the intensity of ester bonds, while texture analysis confirmed a decrease in hardness and an increase in spreadability. The color of the modified samples remained within an acceptable range and therefore did not visually affect the application potential of the fat. From a practical point of view, samples with a higher degree of hydrolysis are the most suitable, showing good properties, especially for cosmetic and food applications. In addition, enzymatic modification allows the gentle processing of animal fats without using chemicals, thus promoting sustainable production and the circular economy. The results of this work open up new possibilities for using previously unused fat by-products, e.g., in food systems and cosmetic matrices.

## Figures and Tables

**Figure 1 molecules-30-03293-f001:**
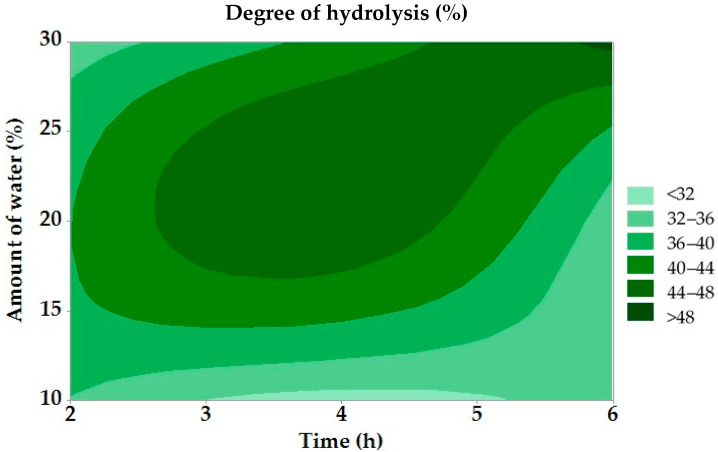
Dependence of degree of hydrolysis on reaction time and amount of water.

**Figure 2 molecules-30-03293-f002:**
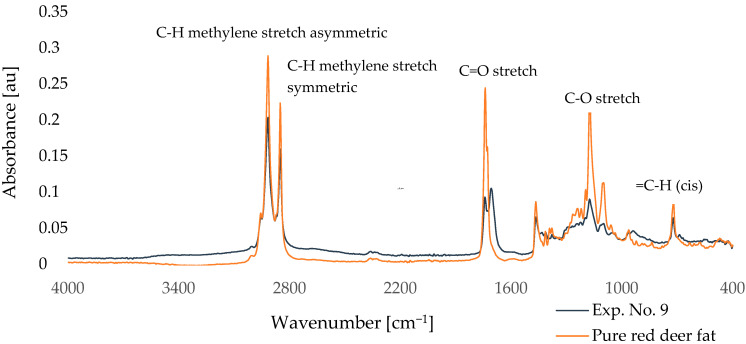
Comparison of FTIR spectra of red deer tallow modified according to Exp. No. 9 and pure red deer tallow.

**Figure 3 molecules-30-03293-f003:**
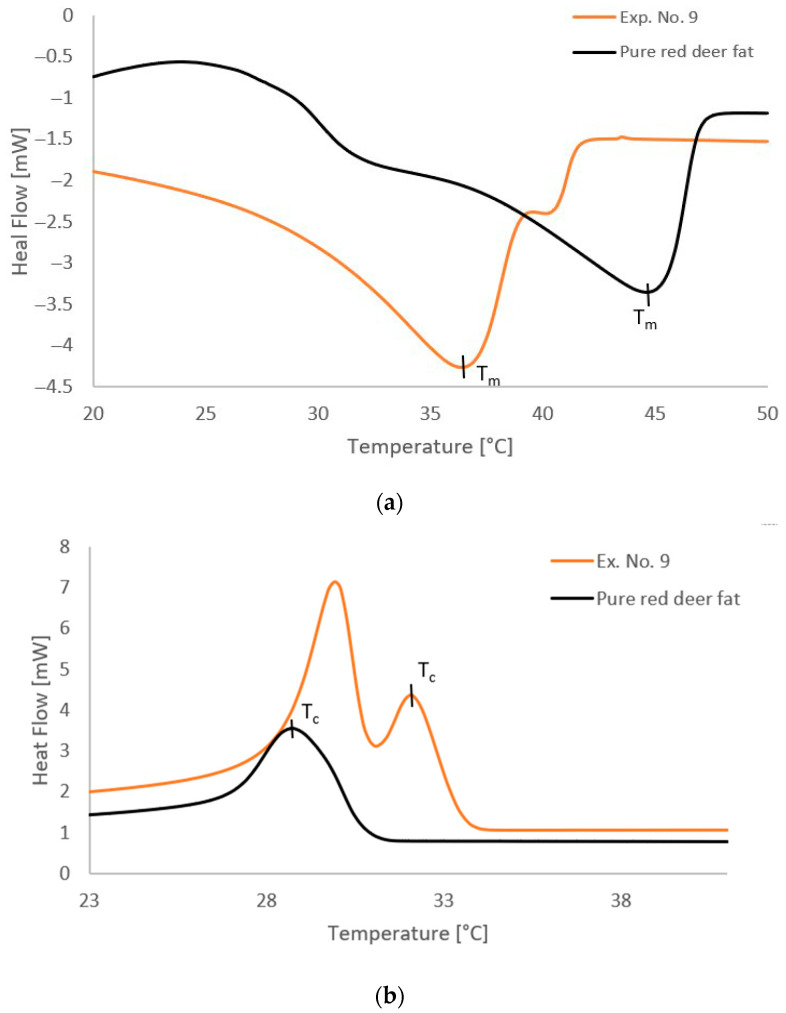
A model run of DSC curves of modified fat according to Exp. No. 9 and pure red deer tallow: (**a**) the melting temperature, (**b**) the crystallization temperature.

**Figure 4 molecules-30-03293-f004:**
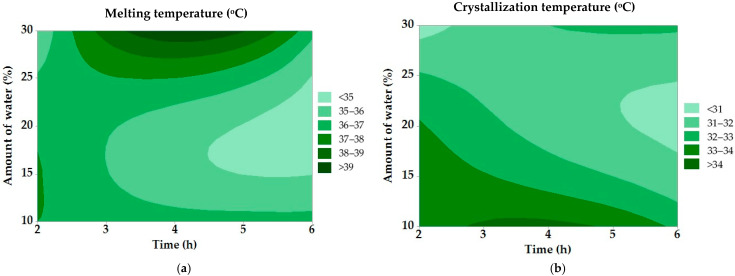
Contour graphs for (**a**) melting and (**b**) crystallization temperature in modified fats.

**Figure 5 molecules-30-03293-f005:**
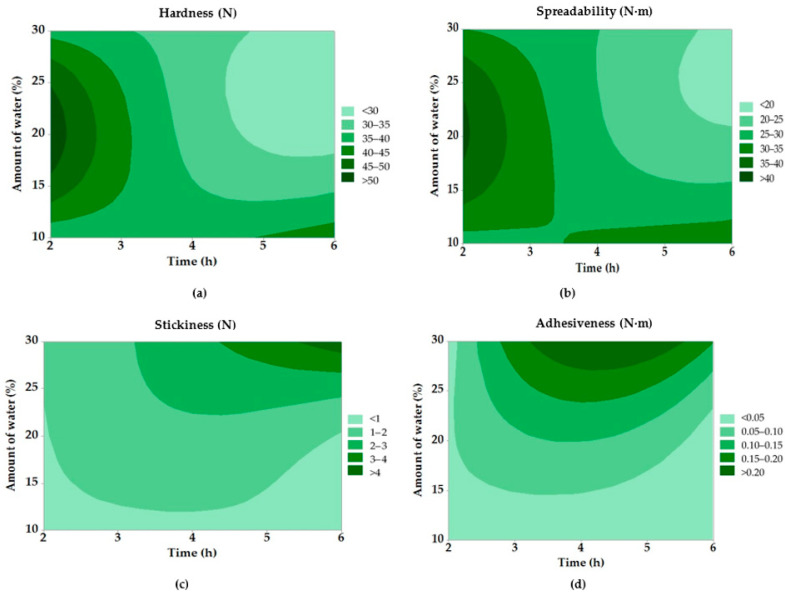
Contour graphs of texture properties of modified fats: (**a**) hardness; (**b**) spreadability; (**c**) stickiness; and (**d**) adhesiveness.

**Figure 6 molecules-30-03293-f006:**
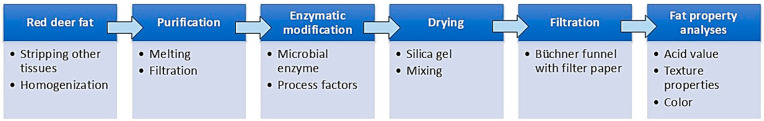
A scheme for the preparation of enzymatically modified red deer fat.

**Table 1 molecules-30-03293-t001:** Fatty acid composition of red deer fat.

Fatty Acid	Amount [%]	Fatty Acid	Amount [%]	Fatty Acid	Amount [%]
C10:0	0.245 ± 0.036	C15:1	6.362 ± 0.001	C18:2 (n6)	21.133 ± 0.057
C12:0	1.305 ± 0.039	C16:0	35.551 ± 0.042	C18:3 (n3)	0.416 ± 0.041
C14:0	7.122 ± 0.011	C16:1	0.349 ± 0.040		
Unknown	0.410 ± 0.008	C17:0	0.993 ± 0.003	Σ SFA	66.64
C14:1	0.956 ± 0.002	C18:0	20.898 ± 0.045	Σ MUFA	11.05
C15:0	0.528 ± 0.002	C18:1	3.734 ± 0.024	Σ PUFA	21.55

SFA—saturated fatty acids, MUFA—monounsaturated fatty acids, PUFA—polyunsaturated fatty acids.

**Table 2 molecules-30-03293-t002:** Process factors of experiments and acid value of modified fats.

Exp. No.	Factor A [%]	Factor B [h]	Acid Value [mg/g]	Degree of Hydrolysis [%]
1	10	2	84.29 ± 1.25	35.79
2	10	4	71.66 ± 2.41	30.43
3	10	6	81.98 ± 0.77	34.82
4	20	2	93.85 ± 0.95	39.85
5	20	4	109.23 ± 1.08	46.39
6	20	6	79.76 ± 0.22	33.87
7	30	2	79.74 ± 0.88	33.86
8	30	4	97.68 ± 0.14	41.48
9	30	6	115.54 ± 1.38	49.07
10 *	20	4	3.41 ± 0.34	1.45
Pure red deer fat	–	–	2.04 ± 0.09	–

Factor A—the amount of water in the reaction; Factor B—the reaction time; *—a blind experiment (without enzymatic treatment).

**Table 3 molecules-30-03293-t003:** An ANOVA table for the results of the DH.

	Degree of Freedom	Sum of Squares	Mean Squares	*p*-Value
Regression equation	DH = 27.85 + 0.389 A + 0.69 B			
Regression	2	102.40	51.20	0.305
Factor A: amount of water [%]	1	91.03	91.03	0.159
Factor B: time [h]	1	11.37	11.37	0.590
Error	6	211.16	35.19	
Total	8	313.56		

**Table 4 molecules-30-03293-t004:** Characteristic peak values of modified fats with reference value and molecular action.

Reference Value [cm^−1^]	700–900	1000–1200	1700–1750	2830–2850	2900–2950
Molecular action	=C-H (cis)	C-O stretch	C=O stretch	C-H methylene stretch symmetric	C-H methylene stretch asymmetric
Wavenumber [cm^−1^]	721	1173	1739	2850	2916
Pure red deer fat	0.088	0.234	0.245	0.224	0.289
1	0.075	0.110	0.104	0.154	0.192
2	0.079	0.119	0.117	0.162	0.203
3	0.073	0.108	0.108	0.159	0.199
4	0.076	0.102	0.098	0.158	0.198
5	0.071	0.094	0.091	0.153	0.191
6	0.078	0.114	0.115	0.163	0.207
7	0.101	0.122	0.118	0.183	0.230
8	0.071	0.104	0.099	0.175	0.225
9	0.056	0.080	0.086	0.149	0.192
10 *	0.090	0.232	0.243	0.221	0.286

*—blind experiment (without enzymatic treatment).

**Table 5 molecules-30-03293-t005:** Melting and crystallization temperatures of modified fats.

Exp. No.	Melting Point Capillary [°C]	DSC—Melting [°C]	Melting Enthalpy [mJ]	DSC—Crystallization [°C]	Crystallization Enthalpy [mJ]
1	36.98 ± 0.31	37.01 ± 0.07	−200.9 ± 1.8	33.36 ± 0.26	52.7 ± 1.4
2	35.10 ± 0.28	36.53 ± 0.17	−174.9 ± 1.0	34.29 ± 0.11	58.0 ± 1.5
3	36.03 ± 0.19	36.42 ± 0.07	−172.7 ± 1.8	32.82 ± 0.01	60.9 ± 0.8
4	37.23 ± 0.08	36.75 ± 0.07	−175.7 ± 0.5	33.11 ± 0.23	44.8 ± 0.4
5	36.48 ± 0.19	35.54 ± 0.02	−213.8 ± 1.5	31.64 ± 0.08	53.7 ± 0.3
6	34.45 ± 0.09	34.53 ± 0.11	−177.9 ± 0.9	30.77 ± 0.02	73.0 ± 0.3
7	36.90 ± 0.16	35.04 ± 0.14	−192.0 ± 0.1	30.53 ± 0.02	49.0 ± 0.5
8	38.00 ± 0.25	39.77 ± 0.09	−248.1 ± 0.7	32.01 ± 0.06	54.9 ± 0.2
9	37.13 ± 0.22	36.31 ± 0.08	−246.8 ± 0.3	32.22 ± 0.03	30.4 ± 0.3
10 *	44.63 ± 0.18	45.07 ± 0.10	−290.6 ± 1.4	29.87 ± 0.02	89.7 ± 0.8
Pure red deer tallow	44.68 ± 0.33	44.52 ± 0.11	−288.1 ± 0.7	28.84 ± 0.24	84.2 ± 0.5

*—blind experiment (without enzymatic treatment).

**Table 6 molecules-30-03293-t006:** ANOVA table for melting and crystallization temperature results of modified fats obtained from DSC measurements.

	Degree of Freedom	Sum of Squares	Mean Squares	*p*-Value
Regression equation	Crystallization = 35.29 − 0.0900 A − 0.250 B			
Regression	2	6.360	3.1800	0.096
Factor A: amount of water [%]	1	4.860	4.8600	0.059
Factor B: time [h]	1	1.500	1.5000	0.243
Error	6	5.369	0.8948	
Total	8	11.729		
Regression equation	Melting point = 36.77 + 0.0083 A − 0.117 B			
Regression	2	0.3683	0.18417	0.945
Factor A: amount of water [%]	1	0.0417	0.04167	0.913
Factor B: time [h]	1	0.3267	0.32667	0.761
Error	6	19.3117	3.21861	
Total	8	19.6800		

**Table 7 molecules-30-03293-t007:** Texture properties of modified fat.

Exp. No.	Hardness [N]	Spreadability [N·m]	Stickiness [N]	Adhesiveness [N·m]
1	35.350 ± 2.159	27.111 ± 4.575	0.585 ± 0.195	0.014 ± 0.008
2	38.149 ± 2.336	30.903 ± 1.826	0.804 ± 0.237	0.020 ± 0.007
3	43.120 ± 3.611	33.900 ± 1.190	0.860 ± 0.228	0.024 ± 0.006
4	52.847 ± 3.394	40.967 ± 3.525	0.935 ± 0.074	0.044 ± 0.020
5	33.376 ± 5.446	25.931 ± 5.992	1.781 ± 0.674	0.101 ± 0.093
6	28.186 ± 5.401	20.740 ± 3.046	0.921 ± 0.479	0.022 ± 0.008
7	37.845 ± 3.215	30.046 ± 1.558	1.025 ± 0.506	0.027 ± 0.009
8	32.564 ± 1.238	25.191 ± 0.702	2.684 ± 1.097	0.249 ± 0.222
9	25.829 ± 1.284	18.129 ± 0.637	4.642 ± 0.986	0.154 ± 0.037
10 *	23.691 ± 2.070	18.452 ± 2.376	1.510 ± 0.291	0.052 ± 0.039
Pure red deer fat	21.384 ± 2.886	16.304 ± 2.353	1.604 ± 0.470	0.074 ± 0.062

*—blind experiment (without enzymatic treatment).

**Table 8 molecules-30-03293-t008:** ANOVA table for texture properties of modified fats.

	Degree of Freedom	Sum of Squares	Mean Squares	*p*-Value
Regression equation	Hardness = 52.78 − 0.338 A − 2.41 B			
Regression	2	207.98	103.99	0.220
Factor A: amount of water [%]	1	68.68	68.68	0.297
Factor B: time [h]	1	139.30	139.30	0.155
Error	6	316.43	52.74	
Total	8	524.41		
Regression equation	Spreadability = 42.74 − 0.309 A − 2.11 B			
Regression	2	164.48	82.24	0.181
Factor A: amount of water [%]	1	57.29	57.29	0.252
Factor B: time [h]	1	107.19	107.19	0.134
Error	6	214.17	35.70	
Total	8	378.65		
Regression equation	Stickiness = −1.72 + 0.1000 A + 0.325 B			
Regression	2	8.535	4.2675	0.054
Factor A: amount of water [%]	1	6.000	6.0000	0.039 *
Factor B: time [h]	1	2.535	2.5350	0.137
Error	6	5.181	0.8634	
Total	8	13.716		
Regression equation	Adhesiveness = −0.0922 + 0.00633 A+ 0.0092 B			
Regression	2	0.026083	0.013042	0.133
Factor A: amount of water [%]	1	0.024067	0.024067	0.061
Factor B: time [h]	1	0.002017	0.002017	0.530
Error	6	0.027206	0.004534	
Total	8	0.053289		

*—statistically significant factor.

**Table 9 molecules-30-03293-t009:** Color parameters of modified fats.

Exp. No.	L*	a*	b*	∆E*
1	90.13 ± 0.29	–3.91 ± 0.03	13.02 ± 0.12	2.90
2	87.17 ± 0.93	–2.65 ± 0.05	13.94 ± 0.35	5.55
3	88.80 ± 0.15	–3.58 ± 0.02	9.11 ± 0.10	2.85
4	88.43 ± 0.17	–3.51 ± 0.06	12.52 ± 0.19	3.66
5	89.07 ± 0.60	–3.76 ± 0.07	10.67 ± 0.34	2.00
6	86.47 ± 0.50	–2.82 ± 0.10	14.20 ± 0.60	5.40
7	85.52 ± 0.27	–0.88 ± 0.10	15.42 ± 0.24	7.61
8	89.07 ± 0.67	–4.16 ± 0.07	13.41 ± 0.69	3.28
9	89.98 ± 0.94	–3.89 ± 0.20	12.33 ± 0.63	2.21
10 *	89.17 ± 0.97	–3.19 ± 0.14	7.33 ± 0.17	3.43
Pure red deer fat	91.56 ± 0.24	–3.70 ± 0.08	11.38 ± 0.16	–

*—blind experiment (without enzymatic treatment).

## Data Availability

The original contributions presented in the study are included in the article; further inquiries can be directed to the corresponding author.

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
