# Peer review of "Enzyme Modifications of Red Deer Fat to Adjust Physicochemical Properties for Advanced Applications"

_molecules, 2025, doi:10.3390/molecules30153293_

Round 1

Reviewer 1 Report

Comments and Suggestions for Authors

The purpose of this manuscript was to study the phase transition, structure, colour, and texture changes of modified fat by enzymatically modifying red deer fat to alter its physical and chemical properties. Overall, the manuscript was interesting. However, there were still some issues that need to be resolved in this work. It could be published after major revision.

  1. Line 55: Why meat with high polar lipid content attracts consumers?
  2. Table 1: What’s the Unknown? Why was it between C14:0 and C14:1?
  3. Figure 3: Whydid only show the DSC spectrum of the No. 9 experimental sample?
  4. Were the reddeer farmed or wild?

Author Response

Manuscript ID: molecules-3756561

Dear Sir / Madam,

Thank you very much for taking the time to review this manuscript. We did our best to revise our paper according to your comments, suggestions, and comments and suggestions from another reviewer.

Please find the detailed responses below and the corresponding revisions and corrections highlighted in red color in the revised manuscript.

According to your suggestion, the figures and tables were improved. Editing of the English language was done as well.

Nine new references were added to the revised manuscript; they are highlighted in red in the Reference list [9, 16, 17, 20, 29, 51, 52, 65, 66].

Yours faithfully,

Pavel Mokrejš (corresponding author)

--------------------------------------------------

Comments 1: Line 55: Why meat with high polar lipid content attracts consumers?

Response 1: Thank you for pointing this out. Meat with a high content of polar lipids, such as venison, attracts consumers primarily due to its enhanced nutritional quality, including the presence of essential fatty acids and phospholipids that are important for human health. Polar lipids, particularly phospholipids, play a crucial role in the structure of cell membranes and have been shown to exert anti-inflammatory and cardioprotective effects. Moreover, they contribute significantly to the sensory attributes of meat—such as flavor and juiciness—which are key factors in consumer perception of quality.

Comments 2: Table 1: What’s the Unknown? Why was it between C14:0 and C14:1?

Response 2: The unidentified peak ("Unknown") listed between fatty acids C14:0 and C14:1 in Table 1 probably represents a less common fatty acid with a comparable chain length, e.g., an isomer, trans form, or methylated derivative that was not detected by the available FAME standard. Such unknown peaks are common in the analysis of natural fats using GC-FID, especially when the sample contains minority fatty acids.

Comments 3: Figure 3: Why did only show the DSC spectrum of the No. 9 experimental sample?

Response 3: Thank you for your comment. The DSC spectrum shown in Figure 3 serves as a representative (illustrative) example and was deliberately selected from experimental sample No. 9, as this sample exhibited the highest degree of hydrolysis (49.07%) and thus showed the most pronounced changes in thermal properties. The intention was to illustrate the contrast between untreated deer fat and the most extensively modified sample, where the changes in melting and crystallization behavior are most evident. Full numerical results for all samples are provided in Table 5 for detailed comparison.

Comments 4: Were the red deer farmed or wild?

Response 4: The red deer used in this study originated from farmed animals. The adipose tissue was obtained in cooperation with Venison CZ Ltd. (Míškovice, Czech Republic), a certified game meat processor sourcing animals from controlled farming operations. This ensured standardized animal handling, traceability, and compliance with hygiene and welfare regulations relevant for scientific sampling.

Reviewer 2 Report

Comments and Suggestions for Authors

This study aims to investigate properties of red deer fat modified by hydrolysis utilizing enzymes. These data is valuable for recycle of scrapped fat. Before publication, minor comments will require author’s responce.

L62-72

 What has the fat contained in deer meat been utilized generally?

If the fat was scrapped, it is better that contents of “SDGs” was inserted in this section.

L161-163

Why is it clarified in this study that cleavage of triacyglycerols occurred?

Is it no possible that the other fatty acid was hydrolysis?

L323-348

Why did short treated fats of number 7 sample have a bigger color change than that of number 8 and 9 samples?

Author Response

Manuscript ID: molecules-3756561

Dear Sir / Madam,

Thank you very much for taking the time to review this manuscript. We did our best to revise our paper according to your comments, suggestions, and comments and suggestions from another reviewer.

Please find the detailed responses below and the corresponding revisions and corrections highlighted in red color in the revised manuscript.

Editing of the English language was done as well.

Nine new references were added to the revised manuscript; they are highlighted in red in the Reference list [9, 16, 17, 20, 29, 51, 52, 65, 66].

Yours faithfully,

Pavel Mokrejš (corresponding author)

--------------------------------------------------

Comments 1: What has the fat contained in deer meat been utilized generally? If the fat was scrapped, it is better that contents of “SDGs” was inserted in this section.

Response 1: Deer fat is often considered a by-product that is discarded or unused, even though it contains essential fatty acids and phospholipids that have potential applications in the food, cosmetics, and pharmaceutical industries (e.g., as emulsifiers or bases for skin and body cosmeceuticals). Enzymatic hydrolysis of animal fats is considered an environmentally friendly technology that enables their conversion into a functional product with high added value. This process supports the circular economy and waste minimization, and also allows for the simultaneous extraction of fatty acids, proteins, and collagen from meat waste. For this reason, the use of deer fat treated by enzymatic methods can be recommended not only as a way of evaluating its potential, but also as a contribution to the achievement of SDG 12 (Responsible Consumption and Production) in the form of valorization of by-products and reduction of the environmental impact of the food industry.

Comments 2: L161-163: Why is it clarified in this study that cleavage of triacyglycerols occurred? Is it no possible that the other fatty acid was hydrolysis?

Response 2: Enzymatic hydrolysis of animal fats is primarily focused on the hydrolysis of ester bonds in triacylglycerols, which are the main component of fat (>95%). Lipase enzymes are completely specific to these molecules and catalyze their cleavage into glycerol and free fatty acids, which were confirmed in the experiment by increased acidity and molecular changes (vibrational characterization of functional groups by FTIR). Given that the study used sn-1,3 microbial lipase, which has high substrate and positional specificity for triacylglycerols, it is highly unlikely that significant hydrolysis of other lipid classes, such as phospholipids, would occur. These enzymes selectively cleave ester bonds at the terminal positions of the glycerol skeleton (sn-1 and sn-3).

Comments 3: L323-348: Why did short treated fats of number 7 sample have a bigger color change than that of number 8 and 9 samples?

Response 3: This phenomenon can be explained by the fact that in sample No. 7, only partial hydrolysis of triacylglycerols occurred, resulting in a larger number of intermediate products (e.g., monoacylglycerols, free fatty acids) that are susceptible to oxidation, thus causing turbidity or affecting the optical properties of the sample. In addition, a short reaction time can lead to incomplete phase separation or higher water retention, which further increases color instability. In contrast, in samples No. 8 and 9, the enzymatic reaction was more profound, the system was chemically more stable, and residual water was better removed, minimizing the possibility of color changes. This effect is also known from other studies, where greater color fluctuations occur in partially modified fat systems.